# Partitioning and Spatial Distribution of Drugs in Ocular Surface Tissues

**DOI:** 10.3390/pharmaceutics13050658

**Published:** 2021-05-04

**Authors:** Anusha Balla, Seppo Auriola, Angus C. Grey, Nicholas J. Demarais, Annika Valtari, Emma M. Heikkinen, Elisa Toropainen, Arto Urtti, Kati-Sisko Vellonen, Marika Ruponen

**Affiliations:** 1School of Pharmacy, University of Eastern Finland, Yliopistonranta 1, 70211 Kuopio, Finland; anusha.balla@uef.fi (A.B.); seppo.auriola@uef.fi (S.A.); annika.valtari@uef.fi (A.V.); emma.heikkinen@uef.fi (E.M.H.); elisa.toropainen@uef.fi (E.T.); arto.urtti@uef.fi (A.U.); kati-sisko.vellonen@uef.fi (K.-S.V.); 2School of Medical Sciences, University of Auckland, Auckland 1142, New Zealand; ac.grey@auckland.ac.nz; 3School of Biological Sciences, University of Auckland, Auckland 1142, New Zealand; n.demarais@auckland.ac.nz; 4Institute of Clinical Medicine, University of Eastern Finland, Yliopistonranta 1, 70211 Kuopio, Finland; 5Faculty of Pharmacy, University of Helsinki, Viikinkaari, 00014 Helsinki, Finland; 6Institute of Chemistry, Saint Petersburg State University, 26 Universitetskii Prospect, 198504 Saint Petersburg, Russia

**Keywords:** cornea, conjunctiva, drug absorption, permeation, beta-blocking agent, MALDI-IMS

## Abstract

Ocular drug absorption after eye drop instillation has been widely studied, but partitioning phenomena and spatial drug distribution are poorly understood. We investigated partitioning of seven beta-blocking drugs in corneal epithelium, corneal stroma, including endothelium and conjunctiva, using isolated porcine tissues and cultured human corneal epithelial cells. The chosen beta-blocking drugs had a wide range (−1.76–0.79) of *n*-octanol/buffer solution distribution coefficients at pH 7.4 (Log D_7.4_). In addition, the ocular surface distribution of three beta-blocking drugs was determined by matrix-assisted laser desorption/ionization imaging mass spectrometry (MALDI-IMS) after their simultaneous application in an eye drop to the rabbits in vivo. Studies with isolated porcine corneas revealed that the distribution coefficient (*K_p_*) between the corneal epithelium and donor solution showed a positive relationship and good correlation with Log D_7.4_ and about a 50-fold range of *K_p_* values (0.1–5). On the contrary, *K_p_* between corneal stroma and epithelium showed an inverse (negative) relationship and correlation with Log D_7.4_ based on a seven-fold range of *K_p_* values. In vitro corneal cell uptake showed a high correlation with the ex vivo corneal epithelium/donor *K_p_* values. Partitioning of the drugs into the porcine conjunctiva also showed a positive relationship with lipophilicity, but the range of *K_p_* values was less than with the corneal epithelium. MALDI-IMS allowed simultaneous detection of three compounds in the cornea, showed data in line with other experiments, and revealed uneven spatial drug distribution in the cornea. Our data indicate the importance of lipophilicity in defining the corneal pharmacokinetics and the *K_p_* values are a useful building block in the kinetic simulation models for topical ocular drug administration.

## 1. Introduction

Topical eye drops are the most commonly used dosage form in the treatment of anterior segment eye diseases [1,2]. Topical drug administration is, however, limited by the low ocular bioavailability [3,4]. For example, 0.07–4.3% of the topically instilled beta-blocking drugs reached the aqueous humor of the rabbit eye [3]. Low ocular bioavailability is due to the short retention time (a few minutes) [5,6,7] and tight permeation barriers of corneal and conjunctival epithelia on the ocular surface [8,9,10]. In fact, a major fraction of the drug dose after topical administration enters the systemic circulation from the ocular surface (via palpebral conjunctiva) and nasal mucosa [11,12,13,14].

In order to be active, the drug needs to reach its target tissue. Topically instilled glaucoma drugs decreasing intraocular pressure should reach trabecular meshwork and anterior uvea (prostaglandins) or ciliary body (beta-blocking drugs, carbonic anhydrase inhibitors) [15]. On the other hand, some clinical drugs act at the ocular surface tissues. These include, for example, cyclosporine in corneal allograft rejection and dry eye disease [16,17,18,19], ganciclovir in ocular viral disease [20], corticosteroids in corneal wound healing [21], ocular surface infections and inflammation (conjunctivitis, keratitis) [22,23]. Drugs should reach adequate concentrations in the target tissues after ocular absorption into and across the cornea and bulbar conjunctiva.

The cornea is divided into three main layers: epithelium, stroma, and endothelium. The epithelium is the outermost layer of the cornea that includes 6–7 layers of nonkeratinized stratified squamous cells (thickness of ~50 µm) with inter-cellular tight junctions in the most anterior layer. Lipoidal epithelium allows permeation of small lipophilic drugs, particularly at Log P ≈ 2–4 [21,24], but its tight junctions limit paracellular permeation of hydrophilic compounds to the size of <2 nm [25]. Acellular and hydrophilic stroma (thickness of ~450 µm) in the middle of the cornea are mostly composed of collagen. The stroma is not a significant diffusional barrier, but the partitioning of lipophilic drugs from the epithelium to the stroma may be slow. The innermost endothelial monolayer does not restrict drug permeation significantly [21,26]. The impact of molecular descriptors (e.g., solubility, molecular size, degree of ionization, hydrogen bonding, lipophilicity) on the corneal drug permeability has been widely studied [21,27,28,29,30,31,32,33,34,35], but spatial drug distribution on the ocular surface and the distribution coefficients between the compartments (epithelium/aqueous vehicle, corneal stroma/epithelium) have not been explored.

The bulbar conjunctiva covers the sclera, and the palpebral conjunctiva lines the inner side of the eyelids. The conjunctiva is composed of the surface epithelium and inner stroma layers. Overall, the epithelium is composed of 2–10 stratified cell layers, including tight junctions [36,37]. The stroma comprises connective tissues with blood and lymphatic vessels. The conjunctiva is more permeable than the cornea [30,38], and its surface area is about 17 times larger than that of the cornea [39]. The conjunctiva may provide a preferred route for the absorption of large and hydrophilic compounds [30,40,41]. A recent study on conjunctival drug permeability [38] demonstrated the key molecular descriptors for conjunctival drug permeation, including polar surface area, hydrogen bond donors, halogen ratio, lipophilicity, and molecular weight. However, the impact of molecular structures on partitioning phenomena in the conjunctival layers has not been determined.

Even though corneal and conjunctival steady-state permeability of drugs has been assessed previously, these permeability coefficients are not applicable in pharmacokinetic modeling of eye drops [42]. After eye drop instillation, ocular drug absorption is characterized by transient drug absorption to the epithelia that takes place in 2–3 min [43], followed by further distribution to the stroma and further. Ocular pharmacokinetic modeling would benefit from a better understanding of the drug distribution between different layers in the cornea and conjunctiva. This study evaluated the partition of beta-blocking drugs with similar molecular weights (248–316 g/mol), but with a range of lipophilicity, to the ocular surface tissues. First, the distribution of beta-blocking drugs was studied ex vivo in porcine corneal epithelium, stroma including endothelium, and the conjunctiva. Second, their corneal distribution was studied in vivo in rabbit eyes with matrix-assisted laser desorption/ionization imaging mass spectrometry (MALDI-IMS) that enables detection of spatial compound distribution in the cornea. Third, the cell uptake kinetics and intracellular concentrations of beta-blocking drugs were studied in cultured human corneal epithelial (HCE) cells.

## 2. Materials and Methods

### 2.1. Drug Molecules

We selected three beta-blocking drugs (propranolol, pindolol, and timolol) for in vivo studies, while seven beta-blocking drugs were used in ex vivo and in vitro studies (Table 1). The stock solutions at 1 mg/mL were prepared from each compound in phosphate-buffered saline (PBS), except pindolol that was dissolved in dimethyl sulfoxide (DMSO; Sigma-Aldrich Chemie Gmbh, Steinheim, Germany). A balanced salt solution enriched with bicarbonate, dextrose, and glutathione (BSS Plus; Alcon Laboratories, South Freeway Fort Worth, TX, USA) was used for further dilution of the stock solutions.

### 2.2. Drug Distribution in Porcine Ocular Tissues Ex Vivo

#### 2.2.1. Preparation of Porcine Tissues

Freshly enucleated porcine eyes were obtained from the local slaughterhouse and were transported in Dulbecco’s phosphate-buffered saline (Gibco, Life Technologies Limited, Paisley, UK). The cornea and conjunctiva were isolated as previously described [38,45].

*Cornea*. A small incision was made at the edge of the limbus through the conjunctiva and sclera into the vitreous, and the cornea was separated from the posterior part of the eyeball by cutting along the limbus. The lens, iris, and ciliary body were peeled off to expose the whole cornea.

*Bulbar conjunctiva.* A 15 mm incision was made along the limbus to separate the conjunctiva from the limbus. A pair of mosquito forceps were inserted into both ends of the cut detaching conjunctiva from the sclera. The conjunctiva, between the two-mosquito forceps, was carefully detached by cutting from sclera, fat, and muscle fibers. Lastly, the piece of the conjunctiva was removed from the eyeball by cutting alongside the forceps.

#### 2.2.2. Drug Distribution into the Ocular Tissues

We used the set-up of permeability experiments in previous studies [10,38]. The isolated corneal and conjunctival tissues were placed on mesh and positioned between two circular silicon rings (aperture 0.64 cm^2^) in the Ussing chambers (Harvard Apparatus, MA, USA). The chambers were filled with pre-warmed (35 °C) BSS Plus supplemented with 10 mM, 4-(2-hydroxyethyl)-1-piperazine ethane sulfonic acid (HEPES; Lonza, Walkersville, MD, USA) (pH 7.4); 5.5 mL in the apical (donor) side and 6.5 mL in the basolateral (receiver) side. Oxygen gas with 5% CO_2_ was supplied to both chambers, and the temperature was maintained at +35 °C throughout the experiment. The transepithelial electrical resistance (TEER) values were measured with glass barrel Ag/AgCl electrodes (NaviCyte Electrodes; Harvard Apparatus, MA, USA), connected to a voltage–current clamp (CC MC6; Physiologic Instruments, San Diego, CA, USA). The TEER values were 337 ± 171 Ω·cm^2^ (*n* = 30) and 105 ± 62 Ω·cm^2^ (*n* = 20) for the cornea and the conjunctiva, respectively.

The experiment was initiated by removing 550 µL of BBS plus buffer from the apical side and adding the same volume containing the cassette mixture of seven beta-blocking drugs (each at a concentration of 2 µg/mL). The samples were taken from the apical side after 10 min (30 µL sample) and at the end of the experiment (500 µL sample).

The tissue samples were collected at the end of each experiment (i.e., 10, 30, 60, 120, and 180 min). The epithelial layer was scraped off with a scalpel from the surface of the cornea, and the stroma containing endothelium was cut into pieces. In the case of the conjunctiva, the tissue inside the circular ring (directly exposed to the drug solution from the donor side) was separated from the remaining tissue on the mesh. All the samples were weighed and stored at −80 °C until liquid chromatography-tandem mass spectrometry (LC-MS/MS) analysis.

#### 2.2.3. Calculation of Partition Coefficients

The estimated partition coefficients (*K_p_*) values were calculated by using the drug concentration values from the distribution study at the 3 h time point. The corneal epithelium/donor and conjunctiva/donor *K_p_* values were calculated as the ratio of beta-blocking drug’s concentration in the tissue (ng/g tissue) and initial concentration of beta-blocking drugs in the buffer at donor compartment (2000 ng/mL). The donor concentration did not change significantly during the experiment. The stroma/epithelium *K_p_* values were calculated as the ratio of beta-blocker concentrations in the stroma (ng/g tissue) and the epithelium. The *K*_p_ values were then analyzed using an ANOVA on ranks test with Dunnett’s test (SigmaPlot 14.0, Systat Software, Inc., San Jose California, CA, USA, www.systatsoftware.com assessed on 21 April 2021).

### 2.3. Cell Uptake in Human Corneal Epithelial (HCE) Cells In Vitro

#### 2.3.1. HCE Cells

Mycoplasma-free immortalized HCE-cells were cultured at +37 °C under an atmosphere of 5% CO_2_ and 95% relative humidity. The growth medium was composed of DMEM (Dulbecco’s Modified Eagle Medium): F12 (1:1 mix; BioWhittaker Lonza, Walkersville, MD, USA) supplemented with 15% fetal bovine serum (FBS 10270, Gibco by Life Technologies, Carlsbad, CA, USA), 0.3 mg/mL-glutamine (EuroClone cat no. ECB3000D, Via Figino, Pero MI, Italy), penicillin–streptomycin solution 1000 U/mL penicillin, 0.1 mg/mL streptomycin (Corning, Ref 30-002-CI, Mediatech Inc., Discovery Boulevard, Manassas, VA, USA), 10 ng/mL epidermal growth factor (Gibco, Thermo Fisher Scientific, Waltham, MA, USA), 5 µg/mL insulin (Gibco, Thermo Fisher Scientific, Waltham, MA, USA), 0.1 µg/mL cholera toxin (Sigma, St. Louis, MO, USA), 0.5% DMSO, and 15 mM HEPES (Fisher BioReagents, Pittsburgh, PA, USA). Upon confluence, the cells were detached with 0.25% trypsin-0.2% EDTA (Gibco, Life Technologies Limited, Paisley, UK) and subcultured 1:10–1:30 twice a week.

#### 2.3.2. Cell Uptake Study

HCE cells were seeded on 24-well plates (1 × 10^5^ cells/well) and were incubated overnight. The next day the cells were washed with pre-warmed buffer (HBSS, pH 7.4) and pre-incubated in the same buffer for 10 min at +37 °C. After pre-incubation, the cells were exposed to the cassette drug solution (30 µg/mL) for 5, 10, 20, 30, and 60 min. After exposure, the drug solutions were collected, and the cells were washed twice with ice-cold HBSS. Thereafter, the cells were lysed by adding 500 µL of 0.1 N sodium hydroxide per well and incubated for 60 min at room temperature. The cells were scraped off, collected, and stored at −80 °C until LC-MS/MS analysis.

#### 2.3.3. Protein Concentration of the HCE Samples

The Bio-Rad protein assay (Bio-Rad Laboratories Inc., Hercules, CA, USA) was used to measure the protein concentration of the HCE cell samples. The bovine serum albumin standards (0.1–2 µg/mL) were prepared in blank cell lysate, and the lysed samples were centrifuged at 2800 rpm for 5 min. The supernatant (5 µL) in duplicates were then transferred into 96-well plates. Dye reagent (Bio-Rad Laboratories, Inc., Hercules, CA, USA) solution (dye reagent concentrate diluted in distilled water at 1:4 ratio) was added to each well (200 µL/well) and incubated for 15 min at room temperature. The absorbance was measured at 595 nm in Victor^2^ multilabel plate reader (PerkinElmer, Wallac, St. Paul, MN, USA)

#### 2.3.4. Determination of Intracellular Drug Concentration

For the determination of the relationship between the cell number and protein concentration, we seeded the HCE cells in cell culture flasks (75 cm^2^) and incubated the cells until they reached 80–90% confluence. The cells were then trypsinized, counted, and the calibration curves (protein concentration vs. cell number) for the range of 75,000 to 400,000 cells/mL were prepared by the dilution of the cells in Eppendorf tubes. The cells were spun down and lysed with 0.1 N sodium hydroxide as described above. Then, protein concentration was measured, and a linear correlation between cell number and protein concentration was determined. The diameter (27.5 ± 5.1 µm) of the cells was measured by an OLYMPUS SLR camera control panel v.2 (OLYMPUS CKX 41, Olympus Life Science, Waltham, MA, USA). The mean volume of HCE cells (1.09 × 10^−8^ mL) was calculated using the following Equation (1).
(1)$$V=\frac{4\pi {\left(\frac{d}{2}\right)}^{3}}{3}$$

Then the total cellular volume was determined by multiplying the volume of a single cell with cell number. The intracellular drug concentrations were calculated by dividing drug amount by total cellular volume in the experimental samples.

#### 2.3.5. Calculation of Partition Coefficients from Uptake Study

We calculated the apparent partition coefficients (*K*_p_) using the drug concentrations of the uptake study at 60 min. The HCE cell/donor *K_p_* values were calculated as the ratio of beta-blocker concentration in the cell lysate (µg/mL) and beta-blocker concentration in the incubation medium (30 µg/mL).

### 2.4. Analyses of Ex Vivo and In Vitro Samples

#### 2.4.1. Sample Preparation for LC-MS/MS

*Corneal epithelial tissue samples.* The corneal epithelial samples were lysed by adding a 38-fold volume of 0.1 N sodium hydroxide, based on the sample weight. The samples were mixed by pipetting and vortexing until homogenous tissue lysate was obtained, and then it was incubated for 15 min at room temperature.

*Corneal stroma and conjunctiva.* The stromal and conjunctival tissue samples were moved to 2 mL tubes containing 2.8 mm ceramic beads (Omni Internationals, Kennesaw GA, USA), and two-fold volume (based on the weight of the sample) of PBS was added to the samples. The samples were processed with Bead Ruptor Elite (Omni Internationals, Kennesaw, GA, USA) for two 4 min cycles at 6 m/s. After the first cycle, s five-fold volume of PBS was added to the samples. The sample was diluted 1:8 during the process.

*HCE cell lysate.* The samples from the exposure solution were diluted 15-fold with HBSS. The cell lysate samples were used without dilution.

*The standards and quality controls (QC)* were prepared from the same drug solution used in the experiments. The calibration standards for the analysis of tissue samples were made in duplicates in the corresponding blank tissue homogenates at the concentrations of 0.5, 1.0, 2.5, 5, 10, 25, 50, 100, and 250 ng/mL. The calibration standards in blank HCE cell lysate homogenates and buffer were prepared at concentrations 0.5, 1.0, 2.5, 5, 10, 25, 50, 100, 250, 500, and 1000 ng/mL. The QCs were independently prepared in blank tissue and cell homogenates at three levels of concentrations 2.5, 25, and 250 ng/mL. An internal standard solution of 240 µL containing 50 ng/mL atenolol-d7 (Toronto Research Chemicals, Toronto, ON, Canada), 5 ng/mL Betaxolol-d5 (Toronto Research Chemicals, Canada), 5 ng/mL timolol-d 5 maleate (Toronto Research Chemicals, Toronto, ON, Canada) in 1% formic acid in acetonitrile was added to an aliquot of 80 μL of samples, calibration standards, or QCs. The samples were vortexed for 1 min, centrifuged at 14,000 rpm for 10 min at 4 °C, and 80 µL of supernatant was transferred into HPLC vials.

#### 2.4.2. LC-MS/MS Analyses

The analyses of samples were performed with LC-MS/MS (Agilent 1290 liquid chromatograph and Agilent 6495 triple quadrupole mass spectrometer, Agilent Technologies Inc., Santa Carla, CA, USA) with an electrospray ionization (ESI) source and multiple reaction monitoring (MRM) mode. Agilent Poroshell^®^ 120 SB-C18 column (2.7 μm, 2.1 × 50 mm) was maintained at 50 °C. Eluent A was 0.1% formic acid (Sigma-Aldrich) in Milli-Q-H_2_O, and Eluent B was methanol (LC-MS Chromasolv, Honeywell, Riedel-de Haen, Seelze, Germany). The eluent flow rate was 0.5 mL/min, and the injection volume 1 µL. At the beginning of gradient elution, 2% mobile phase B was applied for 2 min, then rose to 100% over 5 min for sample elution, followed by column equilibration for 2 min, resulting in a total cycle of 9 min. The precursor and product ions and detailed MS conditions are reported in the Appendix A.

The data were processed and analyzed with Agilent Mass Hunter Quantitative Analysis software (vB.09.00, build 9.0.647.0, Agilent Technologies, Santa Carla, CA, USA). Quadratic fitting with a weighting factor of 1/x^2^ was used in the construction of the calibration curves. The requirements for the acceptance of precision and accuracy were: precision (RSD) should be less than 20%, and accuracy should be within ±20% for ≥67% of calibration and QC levels. The selectivity of ≥ the three-fold response ratio of LLOQ to the matrix-based blank sample was set.

### 2.5. Drug Distribution in Rabbit Cornea In Vivo

Three months old female New Zealand White rabbits, 2.5–2.8 kg (Envigo Laboratories, Blackthron, Bicester, UK), were used in the study to determine the distribution of drugs in the cornea. Eye drops containing 10 mM propranolol, 20 mM pindolol, and 20 mM timolol in PBS (pH 7.2) were applied on rabbit eyes (25 µL/eye). The control animal did not receive any eye drop. The rabbits were sacrificed by injecting a lethal dose (2 mL/kg) of pentobarbital (Mebunat vet 60 mg/mL; Orion Pharma, Finland) into the marginal ear vein at 10, 20, and 60 min post-dosing. The eyes were removed and snap-frozen with liquid nitrogen in plastic tubes and stored at −80 °C until sample preparation and analysis with MALDI-IMS.

#### 2.5.1. Rabbit Tissue Preparation

The frozen eyes were mounted onto a chuck using an optimal cutting temperature compound (Sakura Finetek, Torrance, CA, USA). The samples were sectioned to 20 μm or 35 μm thickness at −20 °C on a Leica CM Cryostat (S3050, Leica Microsystems GmbH, Wetzlar, Germany) and collected on a cryo-film (3C16UF, SECTION-LAB Co. Ltd., Yokohama, Japan) by a modified Kawamoto method [46]. The film was subsequently mounted onto a microscope glass slide using a double-sided copper tape, and the sample slides were dried in a lyophilization chamber [47]. The slides were washed twice with 50 mM ammonium formate for 10 s and dried in a vacuum desiccator. The sample slides contained sections from three eyes from the eye drop experiment and three blanks. The matrix α-cyano-4hydroxycinnamic acid (CHCA) (7 mg/mL in 50% acetonitrile containing 1% trifluoro acetic acid) (Sigma-Aldrich, St. Louis, MO, USA) was added via spray deposition using 10 passes, flow rate of 100 μL/min, temperature of 77 °C, track spacing of 2.5 mm, and velocity of 1300 mm/min (TM-Sprayer, HTX Technologies, Chapel Hill, NC, USA).

#### 2.5.2. MALDI-IMS

Fourier transform-ion cyclotron resonance (FT-ICR) MALDI-IMS was performed using a Bruker 7T SolariX XR mass spectrometer (Bruker Daltonics, Bremen, Germany) at 50 μm spatial resolution for the corneal area and 150 μm spatial resolution for the rest of the eye. Spectra were collected in positive ion mode in a mass-to-charge ratio (*m/z*) range of 150 to 2000, and the mass resolution was 63,000 at *m/z* 786.6. The drugs and *m/z* values ([M + H]^+^) used were as follows: propranolol 260.1645, pindolol 249.1598, and timolol 317.1650. The signal of phosphatidylcholine (PC36:2) at *m/z* 786.599 was collected to be used as a biomarker for cell layers. MALDI images of each signal were plotted with a 5 mDa mass window using flexImaging^TM^ software (v4.1, Bruker Daltonics, Bremen, Germany) with datasets normalized by the RMS method and pixel interpolation.

The relative ionization efficiency of each drug by MALDI was studied by applying standards on different ocular regions. For this study, 35 µm sections of control rabbit eye were collected in an identical manner to those prepared for MALDI IMS. On the tissue sections, 1 µL of the drug mix containing propranolol (5.2 ng), pindolol (9.9 ng), and timolol (12 ng) was spotted in each of the cornea, aqueous humour, and off-tissue. The tissue slides were then washed with ammonium formate and sprayed with the CHCA matrix, as described for the real samples. The spotted regions were then analyzed by MALDI-FT-ICR mass spectrometry, and 50 spectra summed to produce representative spectra for each tissue region to assess ionization efficiency differences.

Since absolute quantitation is not straightforward using MALDI-IMS data, the differential distribution of the drugs in the corneal layers was studied by calculating the normalized signal ratios of pindolol and timolol against twice the signal of propranolol (dose of propranolol was two-fold lower than that of the other drugs). For the drug ratio calculations, the pixels corresponding to the stroma and epithelial layer of the cornea were defined by plotting the MALDI image of a typical cell membrane component, phosphatidylcholine PC36:2 at *m/z* 786.599. The data for calculation was obtained by taking six crosscuts through the cornea; no normalization was used.

## 3. Results

### 3.1. Drug Distribution in Porcine Ocular Tissues Ex Vivo

The distribution of seven beta-blocking drugs was determined in isolated porcine corneal epithelium, corneal stroma, and conjunctiva after drug exposure from the epithelial side in vitro. The lipophilicity of the drugs had an impact on ocular tissue partitioning. For instance, nearly 50 times higher levels of the lipophilic propranolol (Log D_7.4_ 0.79; 9.46 ng/mg tissue) were reached in the corneal epithelium as compared to the concentration of hydrophilic atenolol (Log D_7.4_ −1.76; 0.19 ng/mg tissue) (Figure 1A and Figure 2A). The concentration difference between propranolol and atenolol was about seven-fold (0.83 ng/mg vs. 0.12 ng/mg) in the corneal stroma and about five-fold (1.24 ng/mg vs. 0.27 ng/mg) in the conjunctiva (Figure 1B,C and Figure 2B,C). At the first time point (10 min), the differences between propranolol and atenolol concentrations were lower, ranging from equal (conjunctiva) to 12-fold (corneal epithelium) (Figure 1). The hydrophilic drugs (atenolol, nadolol, and timolol) reached equilibrium faster (≈30 min) than the more lipophilic drugs (e.g., betaxolol, propranolol) (≈60 min) (Figure 1).

The comparison of tissue concentrations revealed that propranolol concentration in the corneal epithelium was 11 and 8 times higher than in the corneal stroma and conjunctiva, respectively. In the case of atenolol, the difference between these tissues was only two-fold. The results indicate that drug distribution on the ocular surface is dependent on the drug lipophilicity and tissue properties (Figure 2).

### 3.2. Apparent Distribution Coefficients in Porcine Ocular Tissues Ex Vivo

We calculated the *K_p_* values for corneal epithelium/donor solution, stroma/epithelium, and conjunctiva/donor solution ratios of the beta-blocking drugs at 3 h (Figure 3). We then determined the relationship between the calculated *K_p_* values with the molecular weight and Log D_7.4_ values. The beta-blocking drugs used in the study have a narrow molecular weight range (Table 1) and did not show any relation to *K_p_* values (Appendix A). The *K_p_* values for epithelium/donor solution showed a positive correlation with Log D_7.4_ values (Figure 4A) as indicated by an ascending trend with increasing *K*_p_ values of beta-blocking drugs from 0.1 atenolol to 4.7 propranolol. The *K_p_* value of propranolol and betaxolol for epithelium/donor solution was significantly higher than the corresponding *K_p_* values of atenolol (*p* < 0.001 and *p* = 0.002, respectively). In contrast, an inverse relationship between *K_p_* of stroma/epithelium was observed with Log D_7.4_ (Figure 4B). The *K_p_* of atenolol for stroma/epithelium was significantly higher by approximately 8 and 4 times than the *K_p_* of propranolol (*p* < 0.001) and betaxolol (*p* = 0.011), respectively (Figure 3 and Appendix A).

In the case of *K_p_* values of conjunctiva/donor solution, a positive correlation was observed (Figure 4C)_,_ but the steepness of the slope is lower (*K_p_* value 0.1–0.4) than in the corneal epithelium/donor solution (*K_p_* values 0.1–4.7). The *K_p_* value of atenolol for conjunctiva/donor solution was significantly lower than that of propranolol and betaxolol (*p* < 0.05; Figure 3 and Appendix A). The *K_p_* values of most beta-blocking drugs were in a similar range for corneal epithelium/donor and conjunctiva/donor, but substantial differences were seen for the lipophilic drugs (betaxolol and propranolol). For instance, in the case of propranolol, the *K_p_* of corneal epithelium/donor was 10 times higher than the *K*_p_ of conjunctiva/donor.

Interestingly, corneal permeability (P_app_,_co_; predicted and experimental) [10,38] correlated poorly with the estimated *K_p_* values but showed a plateau at *K_p_* values > 1 (Appendix A). In the case of the conjunctiva, no plateau was observed, and a positive correlation was observed between conjunctival permeability and conjunctiva/donor solution *K_p_* values (Appendix A).

### 3.3. Uptake of Beta-Blocking Drugs in HCE Cells

The uptake of beta-blocking drugs into the cultured HCE cells was performed using the cassette dosing method. At equilibrium (60 min), the intracellular concentration of lipophilic propranolol was 21-fold higher than the concentration of the hydrophilic atenolol (Figure 5A). The *K_p_* values for the HCE cells/donor at 60 min ranged from 1.6 (atenolol) to 34.4, (propranolol) and was statistically significant (*p* < 0.001; Figure 5B). These results show a similar trend with ex vivo results, and the correlation between *K_p_* of the epithelium/donor solution and *K_p_* of HCE cells for the beta-blocking drugs is very strong (Figure 5C).

### 3.4. Distribution Patterns of the Drugs in the Rabbit Cornea

Drug distribution in the rabbit cornea and aqueous humor in vivo was studied with MALDI-IMS using the overlay of the anatomical structures with drug concentration signals (Figure 6). Propranolol, pindolol, and timolol were detected as protonated molecules that gave signals that were well resolved from the background. The signal intensity of the drugs in the corneal tissues and aqueous humor was higher at 10 min than at 60 min (Figure 6). In the 10 min samples, the most lipophilic propranolol had the highest intensity, followed by pindolol and timolol. At 60 min, timolol was below the limit of detection in many pixels. From the results, we can assume that the drugs were not evenly distributed on the ocular surface, and the levels seemed higher in the superior cornea than in the inferior part.

Analysis of the relative signals was performed by normalizing the signals based on the drug dose (i.e., propranolol signal was doubled as its dose was half of the dose of timolol and pindolol). The relative signals of timolol (Log D_7.4_, −1.43) and pindolol (Log D_7.4_, −0.5) were always below propranolol (Log D_7.4_, 0.76) (Table 2). No substantial ionization efficiency differences were detected between the drugs when they were spotted on the different ocular tissue regions (see Appendix A). Therefore, the ratios of each drug detected in the IMS data could be calculated. Due to the variable nature of MALDI, these calculated ratios were approximate, but the overall trend can be compared with the results obtained with LC-MS. This ratio was higher in stroma than in epithelium, indicating that the concentrations of the drugs were closer to each other in the stroma than in the epithelium. This result was in line with the ex vivo results. Although the ratios did not indicate the true concentrations in the ocular tissues, the relative distribution of the beta-blocking drugs in those ocular tissues can be observed.

## 4. Discussion

In this study, we showed interesting differences in the tissue distribution of seven beta-blocking drugs with varying lipophilicities. The positive and steep relationship of Log D_7.4_ and distribution coefficient (*K_p_*) between the corneal epithelium and donor solution was an expected result since corneal epithelium is a tight and lipoidal membrane that allows absorption of lipophilic drugs (e.g., betaxolol, propranolol) while limiting access of hydrophilic drugs (e.g., atenolol) [46]. This is clearly demonstrated in the studies with de-epithelized corneas: permeability of atenolol increased even up to 23-fold, while permeability of propranolol and betaxolol showed only modest or no changes [29,46]. Cellular uptake studies with cultured HCE cells showed an even better correlation with the *K_p_* values of corneal epithelium/donor solution distribution. Thus, both Log D_7.4_ and HCE cell uptake can predict drug partitioning into the corneal epithelium. It should be noted, however, that Log D_7.4_ as such is not an adequate predictor for corneal permeability [35]. Remarkably, *K_p_* values showed a linear relationship with Log D_7.4_ (Figure 4A) but corneal and HCE barrier permeability values showed sigmoidal dependence on Log D_7.4_ [48]_._ Thus, neither drug partitioning from the tear fluid to the corneal epithelium nor corneal permeability as such can adequately describe corneal drug absorption after eye drop administration.

In contrast to drug partitioning into the corneal epithelium, the relationship between *K_p_* of stroma/epithelium distribution and Log D_7.4_ was negative, indicating an easier distribution of hydrophilic than lipophilic compounds from the cellular epithelium to hydrophilic stroma. Effective absorption to the corneal epithelium and subsequent low distribution to stroma supports the idea that the corneal epithelium acts as a depot for lipophilic drugs [49]. The inverse relationship between Log D_7.4_ and *K_p_* (stroma/epithelium) may explain the sigmoidal relationship between Log D_7.4_ and corneal permeability, thereby giving tools for kinetic modeling of corneal drug absorption as the equilibria of two sequential distribution steps (from eye drop to epithelium; from epithelium to stroma) that can be incorporated into pharmacokinetic models. It should also be noted that in vivo kinetic data suggest that corneal stroma, endothelium, and aqueous humor constitute a joint pharmacokinetic compartment [50].

The conjunctiva is a route for noncorneal ocular drug absorption [51] but is also an avenue for systemic drug absorption from the ocular surface [11]. We observed a positive correlation between conjunctival drug partitioning (*K_p_* conjunctiva/donor) and Log D_7.4_. However, the drug lipophilicity did not seem to affect drug partitioning in the conjunctiva as much as in the corneal epithelium. This is explained by the leakier nature of the conjunctiva than the cornea, thereby allowing easier permeation of hydrophilic molecules [30,38,40,41]. Since porcine conjunctiva utilized in our study contained connective tissue, the values do not represent pure conjunctival epithelium. Nevertheless, our data is consistent with the previous data showing rapid conjunctival systemic absorption of relatively hydrophilic pilocarpine [11] and noncorneal absorption of hydrophilic compounds to the eye [51].

MALDI-IMS was used for the first time to analyze the distribution of several simultaneously applied drugs in vivo in rabbits’ eyes. Previously, only a few MALDI-IMS studies on in vivo ocular pharmacokinetics of atropine [52,53] and brimonidine [54] have been published. In addition, MALDI-IMS was used earlier to analyze the distribution of several drugs in isolated lenses [55] and to detect endogenic metabolites and lipids in the eye [47]. MALDI-IMS is a powerful analytical technique that allows studying the spatial distribution of several drugs in more than one tissue at a time and providing higher spatial resolution than traditional LCMS/MS methods. The signal intensity of timolol and pindolol relative to propranolol were lower in corneal epithelium than in stroma, suggesting that lipophilic propranolol distributes easier to corneal epithelium than to hydrophilic stroma, which is in line with our ex vivo results. MALDI-IMS signals diminished below the detection limit at 60 min after eye drop application because drug retention on the ocular surface in vivo is short.

Interestingly, MALDI-IMS suggested uneven drug distribution on the ocular surface (Figure 6). Previously, the spatial distribution of timolol was studied in vivo in rabbit eyes that were divided into superior and inferior halves [56]. In the cornea, higher concentrations were seen in the half of the eye in which the ocular insert, releasing the drug in a controlled manner, was placed as compared to the other half. In the case of eye drop, the differences were modest, while an inferior drug insert application resulted in 4–6 fold higher timolol levels in the inferior than in the superior cornea. In the current study, the reasons for uneven distribution are unknown but may be related to the kinetics of fluid and drugs in tear fluid as well as structural differences in different regions of the corneal epithelium [57]. The corneal epithelial cells are formed at the limbus, and upon maturation, they migrate toward the center and apical surface of the cornea [58,59]. Continued progress in MALDI-IMS analytics will open unprecedented methodological possibilities for detailed spatial drug analyses that can be overlayed with relevant endogenous factors, thereby offering explanations on regional differences in drug distribution.

## 5. Conclusions

The kinetic parameters, including partition coefficients for corneal and conjunctival pharmacokinetics, were provided. Drug Log D_7.4_ values and in vitro cell uptake can predict drug partitioning into the corneal epithelium. The parameters and their relationship with chemical features provide tools for building pharmacokinetic simulation models that can take into account the transient pharmacokinetics after eye drop instillation. The utility of MALDI imaging mass spectrometry for spatial kinetics of simultaneously administered drugs was demonstrated and showed uneven drug distribution on the ocular surface. Future progress of this powerful method will facilitate understanding of ocular pharmacokinetics and dynamics.

## Figures and Tables

**Figure 1 pharmaceutics-13-00658-f001:**
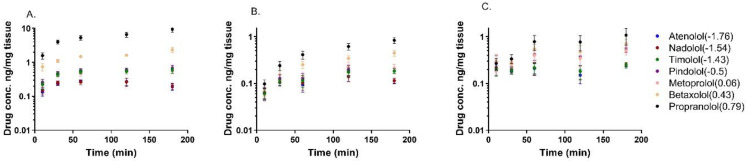
The concentrations of beta-blocking drugs (Log D_7.4_ values are expressed inside parenthesis) in porcine corneal epithelium (**A**), corneal stroma (**B**), and conjunctiva (**C**) as a function of exposure time of drugs to the epithelial side of the isolated tissues. The results are expressed as mean ± standard error of the mean (SEM), *n* = 6 for cornea and *n* = 5 for conjunctiva. The drug concentration in the donor compartment was 2 ng/µL.

**Figure 2 pharmaceutics-13-00658-f002:**
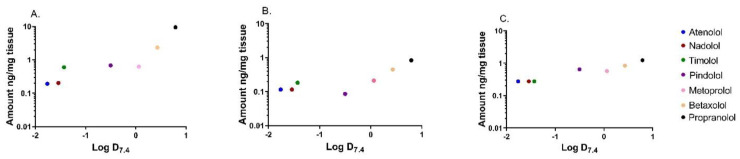
The concentrations of beta-blocking drugs after epithelial side drug exposure ex vivo at 180 min. The concentrations in corneal epithelium (**A**), corneal stroma (**B**), and conjunctiva (**C**) are plotted against Log D_7.4_ of the drugs. The concentration of each drug in the donor compartment was 2 ng/µL.

**Figure 3 pharmaceutics-13-00658-f003:**
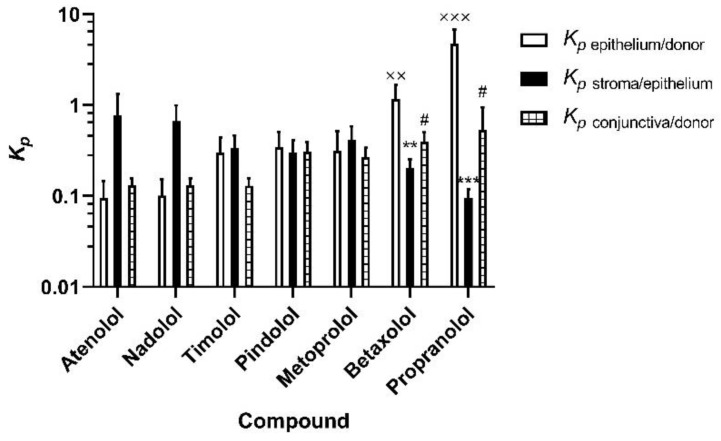
Estimated epithelium/donor solution, stroma/epithelium, and conjunctiva/donor solution distribution coefficients (*K_p_*) for beta-blocking drugs after 3 h drug exposure from the epithelial side. Statistical analysis was done by ANOVA on ranks with Dunnett’s test: ^XX^
*p* < 0.01, and ^XXX^
*p* < 0.001 for *K_p_* epithelium/donor solution, ** *p* < 0.01, and *** *p* < 0.001 for *K*_p_ stroma/epithelium and, ^#^
*p* < 0.05 for *K_p_* conjunctiva/donor solution comparing the values of other beta-blocking agents to atenolol. The results are expressed as mean ± standard error of mean (SEM), *n* = 6 for cornea and *n* = 5 for conjunctiva. Beta-blocking drugs are presented in the order from the most hydrophilic compound (atenolol) to the most lipophilic one (propranolol).

**Figure 4 pharmaceutics-13-00658-f004:**
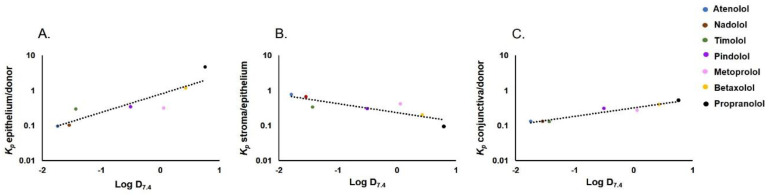
Relationship between Log D_7.4_ and *K_p_* values. (**A**) epithelium/donor, (R^2^ = 0.786), (**B**) stroma/epithelium (R^2^ = 0.732), (**C**) conjunctiva/donor (R^2^ = 0.940).

**Figure 5 pharmaceutics-13-00658-f005:**
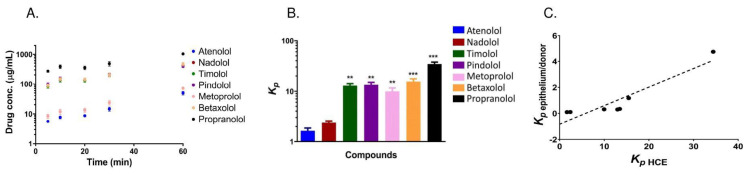
(**A**) The intracellular concentrations of beta-blocking drugs in the HCE cells at different times. The cells were incubated with 30 µg/mL of each drug. (**B**) HCE/donor partition coefficients (*K*_p_) for beta-blocking drugs after 60 min drug exposure. Statistical analysis was done by ANOVA on ranks with Dunnett’s test: ** *p* < 0.01 and *** *p* < 0.001 by comparing the values of other beta-blocking agents to atenolol. The results are expressed as mean ± standard error of the mean (SEM), *n* = 9. (**C**) Correlation between *K_p_* of epithelium/donor and *K_p_* of HCE cells/medium (R^2^ = 0.925). The *K_p_* values represent mean values (*n* = 6 and 9 for *K_p_* epithelium/donor and *K_p_* HCE, respectively.).

**Figure 6 pharmaceutics-13-00658-f006:**
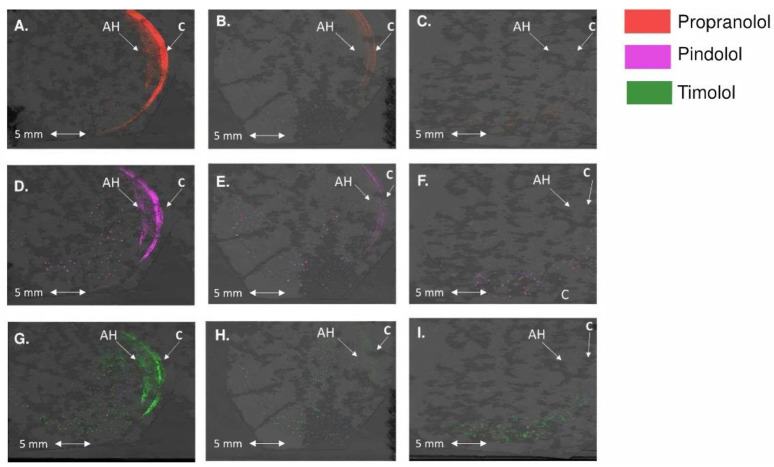
Distribution of (**A**) propranolol at 10 min, (**B**) propranolol at 60 min, (**C**) blank image without propranolol, (**D**) pindolol at 10 min, (**E**) pindolol at 60 min, (**F**) blank image without pindolol, (**G**) timolol at 10 min, (**H**) timolol at 60 min, and (**I**) blank image without timolol after eye drop administration. Drugs were seen in the cornea (C) and aqueous humor (AH). The signals were normalized using the root mean square method.

**Table 1 pharmaceutics-13-00658-t001:** Beta-blocking drugs in the study.

Beta-Blocking Drugs	Log D_7.4_ *	Molecular Weight (g/mol)	Manufacturer
Atenolol	−1.76	266.34	Sigma-Aldrich, St. Louis, MO, USA
Nadolol	−1.54	309.40	Sigma-Aldrich, St. Louis, MO, USA
Timolol	−1.43	316.42	Sigma-Aldrich, St. Louis, MO, USA
Pindolol	−0.5	248.23	Sigma-Aldrich, St. Louis, MO, USA
Metoprolol	0.06	267.36	Sigma-Aldrich, St. Louis, MO, USA
Betaxolol	0.43	307.43	Alcon, Fort Worth, TX, USA
Propranolol	0.79	259.34	Sigma-Aldrich, St. Louis, MO, USA

* [10,44].

**Table 2 pharmaceutics-13-00658-t002:** Ratios of pindolol and timolol signals against twice the signal of propranolol in rabbit corneal epithelium and stroma after eye drop application.

Beta-Blocking Drugs	Epithelium		Stroma	
	10 min	60 min	10 min	60 min
Pindolol	0.21 ± 0.01	0.356± 0.08	0.57 ± 0.00	0.83 ± 0.13
Timolol	0.19 ± 0.02	0.25 ± 0.08	0.36 ± 0.01	0.45 ± 0.17

Note: Data were collected from six to eight replicate regions of interest (ROI) drawn across the cornea. In each ROI, a layer of 300 µm (6 pixels) was averaged, the epithelium was visualized based on its high phosphatidylcholine content. The epithelial layer was typically covered with one or two pixels.

## Data Availability

The data presented in this study are available on request from the corresponding author.

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
