# Peer review of "Partitioning and Spatial Distribution of Drugs in Ocular Surface Tissues"

_pharmaceutics, 2021, doi:10.3390/pharmaceutics13050658_

Round 1

Reviewer 1 Report

The authors have carried out an interesting study on ocular drug absorption after eye drop instillation focusing on partitioning phenomena and spatial drug distribution. The results from this study will be very useful for developing PK models for ocular drug delivery. I feel the results and discussions are presented well and with a few minor improvements, the paper can be accepted.

  1. Figure 1 (and perhaps, figure 5A too!) is difficult to read due to standard error lines interfering between different drugs. Please improve the readability of all the figures.  
  2. In the figure 1 legend, it may be useful to present Log D7.4 values for each drug.
  3. In figure 2A, please recheck the value of Atenolol. From figure 1A at 180 min, I can see it is closer to 0.1 than 1.  
  4. It might be useful if figure 3 is presented using normalized Kp values. For normalization, either Kp for Atenolol or Propranolol can be used.

Reviewer 2 Report

Authors proposed the following paper entitled “Partitioning and spatial distribution of drugs in ocular surface tissues” for the publication in Pharmaceutics, MDPI.

General comments

The paper has a good scientific soundness and deserves to be published after addressing some minor issues.

First of all, an abbreviation list could be added according to the journal guidelines.

Abstract

please define Log D7.4

Line 35. “showed 35

positive correlation with lipophilicity”. may it be “showed low lipophilicity”

line 39. “MALDI-IMS is a promising new technique for future studies on ocular pharmacokinetics 39

and biochemical drug responses” I suggest moving this consideration above, or maybe, eliminate from abstract and moving it to the introduction section

Introduction

Line 61. “cyclosporine A, ganciclovir, corticosteroids, and antibiotics in

the treatment of dry eye disease”. I suggest adding a table listing properties of these drugs and proper references.

line 87. a possible double space is here.

Materials and methods described correctly and deeply. could you add only molecular weights in Table 1?

However, in Line 135, I see “(Ramsay et al., 2017, 2018)”. I think that these references should be mentioned in square parenthesis, as done above. check the guidelines for the insertion of references in the manuscript.

Line 207. this equation should be indicated alone and not inside the text. check the guidelines for equations.

Results

Each of the Figure 6 should contain a reference bar, if possible.

I suggest to reduce Table 2 caption and to leave the  excluded part as comments to the same table.

could you please uniform the number of decimal digits in table 2?

Discussion and conclusion

Line 439. Is “steep” correct in this context?

I suggest developing conclusions section, especially in the aims of future perspectives.

References

Regarding references, I suggest reading and/or adding these papers, concerning the ocular delivery. The reason is that these paper also talk about the use of special drug carriers for ocular delivery

Bouledjouidja, A., Masmoudi, Y., Li, Y., He, W., & Badens, E. (2017). Supercritical impregnation and optical characterization of loaded foldable intraocular lenses using supercritical fluids. Journal of Cataract & Refractive Surgery43(10), 1343-1349.

Campardelli, R., Trucillo, P., & Reverchon, E. (2018). Supercritical assisted process for the efficient production of liposomes containing antibiotics for ocular delivery. Journal of CO2 Utilization, 25, 235-241.

Regarding D7.4 based model, I suggest also:

Systematic Modeling of log D7.4 Based on Ensemble Machine Learning, Group Contribution, and Matched Molecular Pair Analysis DOI: 10.1021/acs.jcim.9b00718
